# Effects of pain education on disability, pain, quality of life, and self-efficacy in chronic low back pain: A randomized controlled trial

Mohammad Sidiq[1], Tufail Muzaffar[2], Balamurugan Janakiraman[3,4], Shariq Masoodi[5], Rajkumar Krishnan Vasanthi[6], Arunachalam Ramachandran[4], Nitesh Bansal[7], Aksh Chahal[1], Faizan Zaffar Kashoo[8], Moattar Raza Rizvi[9], Ankita Sharma[9], Richa Hirendra Rai[10], Rituraj Verma[1], Monika Sharma[1], Sajjad Alam[1], Krishna Reddy Vajrala[1], Jyoti Sharma[1], Ramprasad Muthukrishnan[11] *

1 Department of Physiotherapy, School of Allied Health Sciences, Galgotias University, Greater Noida, Uttar Pradesh, India, 2 Departmet of Physical Medicine & Rehabilitation, Sher-i-Kashmir Institute of Medical Sciences (SKIMS), Srinagar, India, 3 SRM College of Physiotherapy, Faculty of Medicine and Health Sciences, SRM Institute of Science and Technology (SRMIST), Kattankulathur, Chennai, Tamil Nadu, India, 4 Madhav College of Physiotherapy, Faculty of Allied Health Sciences, Madhav University, Rajasthan, India, 5 Department of Endocrinology, Sher-i-Kashmir Institute of Medical Sciences (SKIMS), Srinagar, India, 6 Faculty of Health and Life Sciences, INTI International University Nilai, Nigeri Sembilan, Malaysia, 7 Jindal School of Public Health & Human Development, O P Jindal Global University, Sonipat, Haryana, India, 8 Department of Physical Therapy and Health Rehabilitation, College of Applied Medical Sciences, Majmaah University, Al Majmaah, Saudi Arabia, 9 Department of Physiotherapy, Faculty of Allied Health Sciences, Manav Rachna International Institute of Research Studies, Faridabad, Haryana, India, 10 School of Physiotherapy, Delhi Pharmaceutical Sciences and Research University, New Delhi, India, 11 College of Health Sciences, Gulf Medical University, Ajman, UAE

* mrp@gmu.ac.ae

**Data Availability Statement:** "All relevant data are within the manuscript and its Supporting information files."

## Abstract

### Background

Low back pain stands as a prevalent contributor to pain-related disability on a global scale. In addressing chronic low back pain (CLBP), there is a growing emphasis on incorporating psychological strategies into the management process. Among these, pain education interventions strive to reshape pain beliefs and mitigate the perceived threat of pain. This randomized controlled trial sought to assess the effects of pain education on various aspects, including pain levels, disability, quality of life, self-efficacy, and prognostic characteristics in individuals grappling with CLBP.

### Methods

The clinical trial, retrospectively registered with the Clinical Trials Registry of India (CTRI/ 2021/08/035963), employed a two-arm parallel randomized design. Ninety-two participants with CLBP were randomly assigned to either the standard physiotherapy care with a pain education program or the control group. Both groups underwent a 6-week intervention. Assessment of pain intensity (using NPRS), disability (using RMDQ), self-efficacy (using the general self-efficacy scale), and well-being (using WHO 5I) occurred both before and after the 6-week study intervention.

**Funding:** The author(s) received no specific funding for this work.

**Competing interests:** The authors declare that they have no known competing financial interests or personal relationships that could have appeared to influence the work reported in this paper.

## Findings

Post-intervention score comparisons between the groups revealed that the pain education intervention led to a significant reduction in disability compared to the usual standard care at 6 weeks (mean difference 8.2, $p < 0.001$, effect size Cohen d = 0.75), a decrease in pain intensity (mean difference 3.5, $p < 0.001$, effect size Cohen d = 0.82), and an improvement in the well-being index (mean difference 13.7, $p < 0.001$, effect size Cohen d = 0.58).

## Conclusion

The findings suggest that integrating a pain education program enhances the therapeutic benefits of standard physiotherapy care for individuals dealing with chronic LBP. In conclusion, the clinical benefits of pain education become apparent when delivered in conjunction with standard care physiotherapy during the management of chronic low back pain.

## Introduction

Chronic low back pain (CLBP) is a pervasive and debilitating condition affecting millions globally. It poses a substantial global health burden, inducing significant pain, disability, and diminishing the quality of life for those afflicted. The prevalence of CLBP resulted in a staggering total of 60.1 million person-years living with disability [1, 2]. Lifestyle factors, such as physical inactivity, excess weight, poor sleep, and smoking, contribute significantly to the burden of low back pain (LBP). Particularly noteworthy, several studies identify smoking habits as a crucial prognostic predictor influencing CLBP-related interventional outcomes [3, 4]. Recent trials have recommended exploring non-clinical factors predicting prognosis to enhance the understanding of interventions related to CLBP [5]. To address the multifaceted nature of CLBP, comprehensive treatment approaches are imperative, surpassing solely pharmacological interventions or a combination of pharmacological and physical rehabilitation [6, 7]. Various non-surgical interventions have been implemented to alleviate CLBP, encompassing techniques such as joint manipulation, acupuncture, traditional and contemporary therapeutic exercises, electrotherapy modalities, and medication [8].

However, these interventions exhibit limited to no efficacy in addressing psychological barriers in the recovery from low back pain [9]. Psychological factors play a significant role in an individual's experience of LBP, impacting their function, pain perception, self-efficacy beliefs, and overall quality of life [6, 9, 10]. In the current landscape of multimodal interventions for the management of CLBP, there is a growing recommendation for approaches that involve pain education programs delivered alongside standard physiotherapy care [11, 12]. Pain education, as a psychological approach, centers on enhancing knowledge and understanding of pain. It employs biological and neurophysiological-biomechanical explanations to facilitate the reconceptualization of beliefs surrounding the experience of pain, particularly in chronic scenarios [13].

Despite the increasing recommendation to incorporate pain education in addressing chronic LBP, a notable challenge arises from the variability in the effectiveness of pain education content. This variability is influenced by factors such as the complexity of the curriculum, ethno-cultural considerations, individual pain experience context, and language, all warranting thorough examination [11, 12, 14]. Pain education interventions aim to empower individuals by providing them with knowledge and tools to better comprehend their pain experience.

The overarching goal is to reduce pain-related disability, enhance the quality of life, and boost self-efficacy in managing their condition. The introduction of this form of pain education, often referred to as 'Pain Neuroscience Education' or 'Explain Pain' [15, 16], has been embraced by several Western countries, yielding mixed findings regarding its efficacy. Further, existing literature [11, 14, 17, 18] indicates that psychological approaches proven effective in one cultural context may not necessarily translate to similar efficacy in another. There is a notable dearth of efforts aimed at evaluating or adapting pain education materials, as well as implementing pain education programs, within the Eastern cultural context, particularly in South Asia [18, 19].

Although pain education is endorsed for CLBP by several Western studies [20–24], there arises a critical need to develop and test curricula specifically tailored to the Eastern cultural context. Conversely, the impact of psychological approaches on chronic pain is heavily contingent on patients' educational, ethnocultural, and social backgrounds. This dependency not only limits the generalizability of existing literature results to different contexts but also underscores the necessity for multicultural, contextual investigations in conditions involving pain catastrophizing, such as LBP. In light of these considerations, we hypothesize that patients subjected to pain education will exhibit significant improvements in disability, pain levels, quality of life, and self-efficacy compared to those in the control group with chronic low back pain among individuals with CLBP in India. The primary objective of this study is to assess whether pain education, delivered through a structured intervention, results in a notable reduction in disability and pain intensity compared to a control group receiving standard care. The secondary objectives aim to determine the impact of pain education intervention on the quality of life, self-efficacy, and prognostic characteristics of individuals with CLBP.

## Methods

### Study design and ethical consideration

This randomized controlled trial employed a 2-arm parallel-group design with a 1:1 allocation ratio, utilizing a block size of 4. Random sequences within each block were generated using index cards folded in opaque envelopes for randomization [25]. The study adhered to the Standard Protocol Items; Recommendations for Interventional Trials [26] statement during the protocol development (S1 File) and followed the CONSORT 2010 [27] guidelines (Fig 1). The trial protocol received approval from the Institutional Ethics Committee of SKIMS under the reference RP/114/2021 and was registered with the Clinical Trial Registry of India (CTRI/2021/08/035963). There were no modifications to the methods after the trial registration. All methods in this trial were conducted in accordance with the Declaration of Helsinki. Participants were fully informed about the study's purpose and aim, with emphasis on the voluntary nature of participation and the right to withdraw at any point. After the recording of post-intervention outcome measures for the control group participants, the pain education manual was provided to them along with a brief 10-minute lecture.

### Participants and study setting

This study was conducted as a prospective parallel-group active-controlled trial from September 2021 to October 2022 in the Department of Physical Medicine and Rehabilitation (PMR) at Sher-e-Kashmir Institute of Medical Sciences (SKIMS), a tertiary public healthcare facility in Srinagar, Kashmir, India. Screening of eligible participants continued until the required sample size of 92 patients was attained. Inclusion criteria comprised participants diagnosed with nonspecific low back pain lasting more than 3 months, as diagnosed by physiatrists through clinical examination and diagnostic procedures, aged between 18 to 60 years, of both

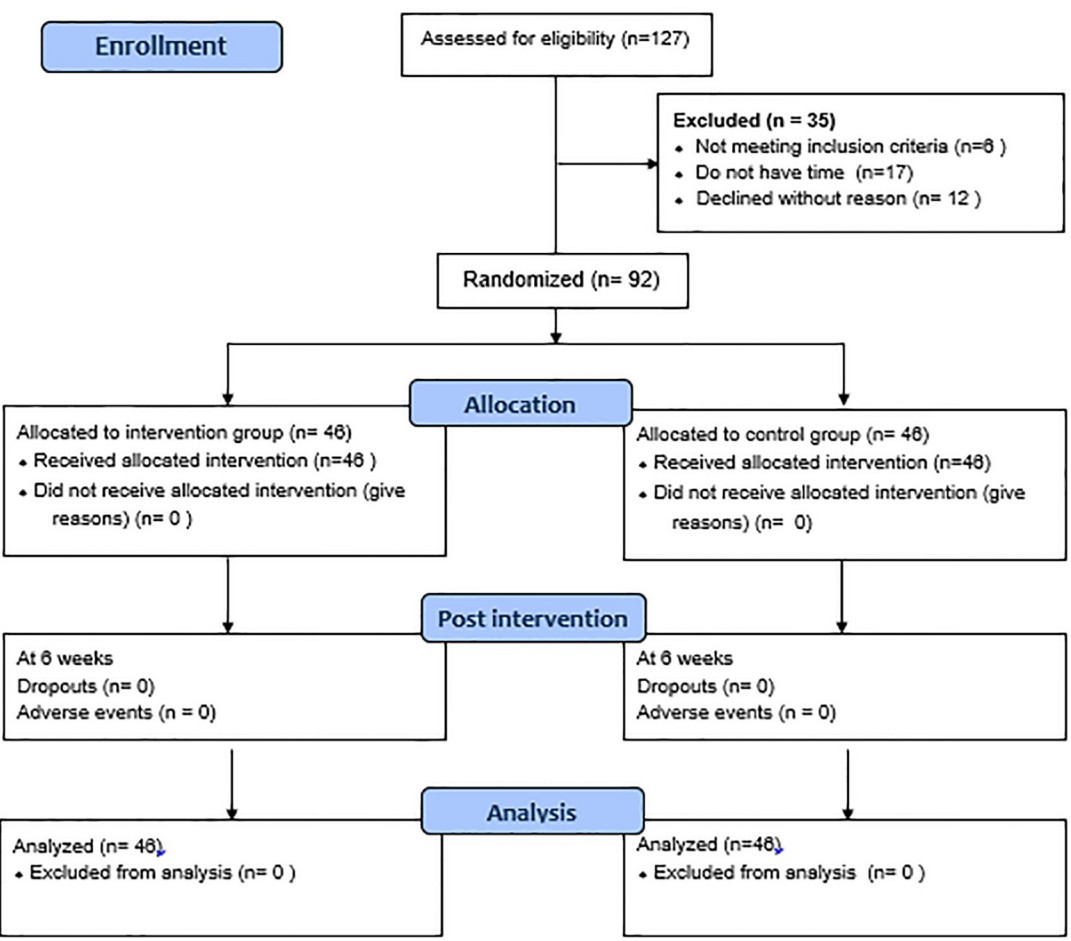

**Fig 1. CONSORT flow diagram of the study.**

genders, and able to attend all intervention sessions at the study setting. Exclusion criteria included voluntary withdrawal from the study for any reason and missing more than two intervention sessions. The participants were blinded to their group assignment, as were the physiotherapists who conducted the clinical outcome examinations. All participants were referred by PMR physicians and were apprised of the voluntary nature of participation; informed consent was obtained from each participant. This trial adhered to the guidelines outlined in the Helsinki Declaration [28].

## Sample size calculation

The required power-calculated sample size was determined using the following assumptions [29]: a confidence interval of 95%, aiming for a power of 0.80 (80%), and an α error probability

of 0.05 (two-tailed test), based on the therapy effects on pain intensity observed in a previous study (Cohen d = 0.595) [30], for the statistical test (t-tests) of difference between two independent means (two groups). The mean scores of pain intensity among chronic LBP patients were utilized (mean1 = 3.76, mean2 = 4.78, SD1 = 1.51, SD2 = +1.91). Consequently, the G power software version 3.1.9.4 for Windows estimated a sample size of 46 participants per group.

## Randomization and blinding

A concealed allocation method utilizing sequentially numbered, opaque, and sealed envelopes (SNOSE) [31] was employed to randomly allocate participants to either the pain education group or the standard physiotherapy care group. Random allocation was executed by the office clerk of the PMR department through block randomization, utilizing cards labeled A and B, with a block size of 4. Participants remained unaware of their group assignment, and both the assessors and the data analyzer of the outcome measures were blinded to the intervention group assignment.

## Description of interventions

Both groups of participants received guideline-based physical therapy intervention extracted from clinical practice guidelines for CLBP [32, 33]. The treatment regimen included advice to avoid bed rest, superficial heat application (10 minutes), lumbar region musculature stretching (10 minutes), static cycling (10 minutes), and core exercises targeting musculature strength and endurance (10–15 minutes). Additionally, the pain education group underwent a face-to-face education program consisting of modules addressing chronic pain definition, pain neurophysiology, and neurobiological aspects, including central sensitization, fear avoidance factors, and social factors influencing low back pain experiences. The content of the pain education was delivered by the first author to individual participants using PowerPoint materials, images, lectures, and question-and-answer sessions, spanning two sessions per week for the first three weeks. The 4th and 5th weeks included question-and-answer sessions reflecting on the learning, and at the conclusion of the 6th session, a brief pain education manual was provided to participants in the pain education group. On average, routine physiotherapy care lasted 40 minutes for both groups. Adherence to both physiotherapy and pain education interventions was recorded using a study calendar during face-to-face education program sessions. Treatment fidelity of the pain education sessions was monitored and ensured through the delivery steps by T.M and S.M.

## Pain education manual development

A context- and culture-specific pain education manual was developed in Hindi by the authors MS and BJ, following the process recommended by Butler and Moseley [15, 16, 18]. Co-authors (SM, AC, FZK, NB) reviewed the Hindi pain education manual to ensure clarity and simplicity of the contents. The authors utilized literature related to pain education, clinical guidelines for treating LBP, pain narratives in Hindi to elucidate target pain concepts, and the pain education handbook (Explain Pain), which significantly contributed to the development of the Hindi pain education manual. The final proofreading of the manual (S2 File) was conducted by a clinical psychologist, a physiatrist, a native Hindi speaker, and a physiotherapist at SKIMS.

## Outcome measures

The socio-demographic characteristics of the study participants were recorded following the recommendations of the National Institute of Health (NIH) Task Force on Research Standards

for Chronic LBP [34]. The study outcomes comprised self-reported tools selected to assess participants' pain intensity, disability related to LBP, self-efficacy, and well-being. These outcomes were measured both at baseline and at the end of the 6-week intervention period.

**Primary outcome measures.** The primary outcome measures focused on assessing the intensity of pain experienced during activities of daily living (ADL) and the disability associated with pain. Pain intensity was evaluated using the Visual Analogue Scale (VAS), with scores ranging from '0' (no pain) to '10' (worst imaginable pain), where participants marked their perceived pain intensity on the scale during ADL. Disability-related to pain was assessed using the 24-item Roland Morris Disability Questionnaire (RMDQ), with scores ranging from '0' (no disability) to '24' (high disability) [35].

**Secondary outcome measures.** The secondary outcome measures included assessments of participants' belief in their ability to cope with and succeed in challenging situations using the General Self-efficacy Scale (GSE), as well as evaluations of emotional well-being using the World Health Organization Five Well-Being Index (WHO-5) [36, 37]. The GSE is a self-reported measure consisting of 10 items, with scores ranging from '10' to '40', where higher scores indicate greater self-efficacy. This scale evaluates individuals' optimistic self-beliefs regarding their ability to cope with various life demands. The WHO-5 is a self-reported measure comprising 5 items assessing current mental well-being. Raw scores on the WHO-5 index range from 0 to 25, with '0' indicating the worst possible quality of life and '25' indicating the best possible quality of life. The raw score is then multiplied by 4 to compute the percentage of the WHO-5 index representing quality of life (percentage score range 0–100). In this trial, the WHO-5 index was expressed as a percentage of quality of life.

## Data analysis

The patient characteristics at the baseline of this pain education intervention trial are presented as mean (SD) for continuous variables or frequency with percentage for categorical variables. In addition, student's t-tests and Chi-square tests were used to evaluate the statistical difference between groups at baseline. The normality of the score distribution of all the outcome measures was tested with the Kolmogorov-Smirnov test, and Levene's test was used to check the assumption of homogeneity. Unadjusted mean (SD) and standard error (SE) of the mean difference (MD) between the groups with associated 95% confidence interval and p-value were computed for the outcome measures at baseline and post-intervention using linear mixed models with unstructured covariance structure adjusting for the baseline value, sex, age, and smoking, and by using Student's t test. Literature reported potential prognostic variables such as educational status, BMI, smoking, chronicity of symptoms (months), pain at baseline, disability (RMDQ) at baseline, WHO 5 Index, and GSE at baseline were included in the prognostic model. We performed a prognostic model logistic regression analysis by categorizing (dichotomous) the post-intervention scores at the closest value to the 75th percentile (to represent 25% of those who did not improve versus 75% representing those who improved) [38]. The Post-intervention RMDQ score was dichotomized at $\geq$ 13, VAS at $\geq$ 5, WHO-5 well-being index at $\geq$ 60, and GSF at $\geq$ 33. These cutoffs are designed to differentiate LBP patients who considered themselves not recovered versus those with high-level recovery, and this method is recommended in the literature in the absence of meaningful cut-off. The level of significance was set at 0.20 and 0.05 for univariate and multivariate logistic regression, respectively, to determine the main effect of potential prognostic variables (independent variables) on the outcome variable. Limiting entry to a few variables identified in the univariate analyses is the most recommended approach and it minimizes the count of variables until the most parsimonious model is identified. This approach also enhances numerical stability and

generalizability of the results. Further, the potential significant independent variables included in multivariate model will interact with other variables, simultaneously adjusting for the other incorporated variables in model. The strength of the association between the potential predictors and outcome measures were expressed as odds ratio (OR). The effect size of the intervention was calculated as the mean difference using Cohen's d effect size. Cohen's d was utilized to assess the clinically important difference, with an effect size of 0.2 indicating a small effect, 0.5 denoting a moderate effect, and 0.8 representing a large effect [39]. The level of significance was set at 5%. All the analyses were based on the intention-to-treat analysis. Analyses were performed using SPSS version 26 for Windows as per the prespecified statistical analysis plan that was proposed during protocol draft (S3 File).

## Results

### Baseline characteristics

92 respondents diagnosed with chronic LBP through clinical examination or imaging agreed to participate in this trial, and none dropped out until the completion of the intervention schedule period. Recruitment took place from 3rd September 2021 to 9th October 2022 prospectively. Table 1 illustrates the baseline characteristics of the experimental and control group participants. The mean age of patients in the experimental group was 42.0 ± 8.2 years, while in the control group, it was 42.5 ± 13.3 years. The variables showed statistical similarity between the experimental and control groups at baseline, except for smoking habits and RMDQ scores. In the occupation subgroup, we observed a higher proportion of housewives (63%) in the control group, while the majority (47.8%) of participants in the experimental group self-reported being self-employed. The primary outcome measure (RMDQ) exhibited a statistically significant difference between the groups at baseline. The experimental group had a better mean score for RMDQ (RMDQ = 15.02, p 0.019, mean difference (MD) = 2.37), and also had a higher number of smokers compared to the control group (n = 11 (23.9%) versus n = 4 (8.7%), p 0.044). Participants in both groups self-reported statistically similar pain intensity of above 5 out of 10 on VAS at baseline (control group 5.98 and experimental group 5.7). The mean duration of chronicity of LBP among the participants was 7.4 months for the control group and 7.8 months for the experimental group, respectively. The secondary outcome measures WHO5 index and GSE showed no statistically significant differences between the groups at baseline (see Table 1). The baseline assessment contained two missing items: one item (GSF) for each of the two participants and one item (WHO5I) for one participant. The respondents' mean score for each item answered to each measure was used to fill in the missing values for the WHO5I and GSF items.

### Intervention findings

**Primary outcomes.** *Disability*. The patient education intervention, using a structured booklet, appeared to significantly reduce disability, as measured by RMDQ, and intensity of pain, as measured by VAS. The change in RMDQ scores within both groups from baseline to post-intervention was statistically significant (Experimental group: mean difference (MD) 6.8, 95% confidence interval (CI) 0.4, 0.6, p < 0.01; Control group: MD 1.0, 95% CI 0.6, 1.3, p < 0.01) (see Fig 2).

The between-group mean difference in RMDQ score change at the post-intervention was 8.0 (p < 0.001), with a large effect size (Cohen's d = 0.75). The clinical change of RMDQ over time (6 months) was computed using the formula ((baseline score minus post-intervention (MD)) / baseline score) x 100. For the experimental group, at the start of the intervention, the RMDQ score was 12.65, and the immediate post-intervention score was 5.8, indicating an

**Table 1. Baseline characteristics on demographic factors, behavioural characteristics, clinical charaterisitics, and study outcome measures (n = 92).**

| Variables | Control (n = 46) | Experimental (n = 46) | P |
|---|---|---|---|
| Age (years) | 42.5 ± 13.3 | 42.0 ± 8.2 | 0.829 |
| Gender (male/female)[†] | 11(23.9)/35(76.1) | 20(43.5)/26(56.5) | 0.077 |
| Occupation [†] | | | 0.531 |
| Self-employed | 14 (30.4) | 22 (47.8) | |
| Housewife | 29 (63.0) | 15 (32.6) | |
| Employee | 0 | 2 (4.3) | |
| Student | 3 (6.6) | 7 (15.2) | |
| Duration of LBP in months | 7.4 ± 1.2 | 7.8 ± 1.3 | 0.768 |
| Level of activity [†] | | | 0.681 |
| Very active | 9 (19.6) | 8 (17.4) | |
| Moderate | 23 (50.0) | 20 (43.5) | |
| Light | 14 (30.4) | 18 (39.1) | |
| Height (m) | 1.58 (0.07) | 1.56 (0.06) | 0.112 |
| Weight (kg) | 66.84 (6.8) | 66.04 (3.3) | 0.474 |
| BMI (kg/m2) | 26.9 (4.0) | 27.8 (4.2) | 0.858 |
| Comorbidity [†] | | | 0.314 |
| None | 33 (71.7) | 37 (80.4) | |
| Hypertension | 4 (8.7) | 5 (10.9) | |
| Diabetes mellitus | 9 (19.6) | 4 (8.7) | |
| Smoker [†] | | | |
| Yes | 4 (8.7) | 11 (23.9) | 0.044* |
| No | 42 (91.3) | 35 (76.1) | |
| Hours of sleep | 6.3 (1.7) | 6.7 (1.3) | 0.549 |
| Pain intensity (VAS 0–10) | 5.98±1.7 | 5.7±1.65 | 0.423 |
| Disability RMDQ (0–24) | 15.02±4.7 | 12.65±4.7 | 0.019 |
| WHO5I (0–100) | 54.4±18.6 | 53.5±17.5 | 0.804 |
| GSF (10–40) | 25.78±6.4 | 25.82±5.6 | 0.972 |

[†]Categorical variables are expressed as frequency and percentage, continuous data as mean and ± Standard deviation, [†] Chi-square test, LBP: low back pain, RMDQ: Roland-Morris Disability Questionnaire, WHO5I: WHO-5 Well-being Index, GSF: General Self-efficacy Scale.

* indicates statistical significance.

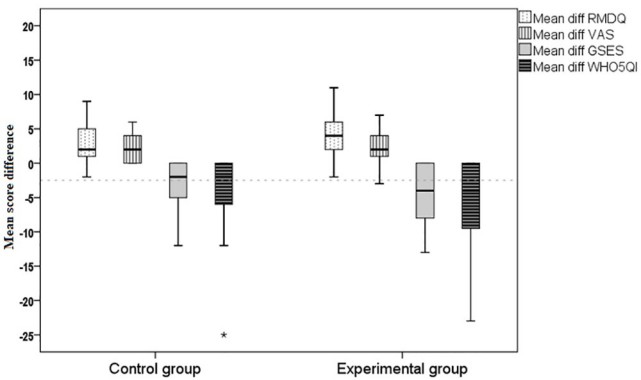

**Fig 2. Mean difference of pre-test and post-test scores of RMDQ, VAS, GSF, and WHO5QI in control and experimental groups.**

improvement of 6.85 points or a 54.15% decrease in the level of disability. Conversely, the mean RMDQ score of the participants in the control group was 15.02 at the start of the intervention and 14.0 immediately after the intervention period, reflecting a clinical change over time (6 weeks) of 1.02 points improvement or 6.8%. The pain education program led to a notable and statistically significant decline in disability related to LBP, as reported by participants in the experimental group at 6 months post-intervention (see Table 2).

*Pain.* The intensity of pain was measured using the self-reported VAS, a tool that can be completed in less than 1 minute. The scale consists of a 10 cm horizontal line labeled "no pain" on the left (assigned '0') and "worst imaginable excruciating pain" on the right (assigned '10'). Pre-/post-test scores of pain (VAS) between the experimental and control groups are shown in Table 2. Pain significantly alleviated within both the experimental (mean difference 2.20, 95% CI; 1.55, 2.85, $p < 0.01$) and control groups (mean difference 2.435, 95% CI; 1.87, 2.99, $p < 0.01$), with a significant main effect and large effect size ($p < 0.001$, Cohen's d = 0.82) between the two groups at post-intervention.

*Secondary outcomes.* The WHO-5 well-being index differed significantly between baseline and 6 months of intervention in favor of the pain education group, with a moderate effect size ($p < 0.001$, Cohen's d = 0.58). However, no significant difference was found for the GSE scale (Table 2).

*Regression model for prognostic predictors.* In addition to testing the hypothesis on the efficacy of the intervention, we determined the association of potential prognostic variables using a logistic regression model (Table 3). Self-reported higher RMDQ at baseline increased the odds of elevated disability at 6 weeks (OR 1.99, 95% CI 1.42, 3.86), while a higher educational status of the participants was protective against higher disability at post-intervention (OR 0.75, 95% CI 0.41, 0.89). A higher VAS score at baseline was associated with 1.69-fold increased odds of higher pain intensity at 6 weeks (OR 1.69, 95% CI 1.31, 2.03). Better well-being and

**Table 2. Unadjusted means, standard deviations, standard error, and 95% CI of the mean difference of the continuous outcome measures for control and experimental group (n = 92).**

| Outcome measures | Experimental (n = 46) | Control (n = 46) | p-value (ES cohen d) |
|---|---|---|---|
| VAS | | | |
| Post intervention | 2.21 (0.9) | 5.6 (1.15) | p <0.001 (0.82)[a] |
| Improvement (95%CI of MD) | 3.6 (3.2, 4.0) | 0.34 (0.2, 0.4) | |
| SE of mean | 0.18 | 0.87 | |
| RMDQ Score | | | |
| Post intervention | 5.8 (3.4) | 14.0 (4.6) | p <0.001 (0.74)[a] |
| Improvement (95%CI of MD) | 6.8 (0.4, 0.6) | 1.0 (0.6, 1.3) | |
| SE of mean | 0.40 | 0.15 | |
| WHO-5 Score | | | |
| Post intervention | 70.1 (14.7) | 56.4 (17.8) | p <0.001 (0.58)[a] |
| Improvement (95%CI of MD) | 17.1 (14.3, 20.0) | 2.0 (1.1, 2.8) | |
| SE of mean | 1.41 | 0.41 | |
| GSF | | | |
| Post intervention | 28.8 (5.4) | 30.22 (4.36) | p 0.097 (-0.035)[a] |
| Improvement (95%CI of MD) | -3.04 (-4.11, -1.98) | -4.39 (-5.6, -3.09) | |
| SE of mean | 0.52 | 0.64 | |

VAS-Visual analog scale; RMDQ-Roland Morris Disability Questionnaire; WHO-5-The World Health Organization- Five Well-Being Index, GSF: General Self-efficacy Scale, CI of MD—Confidence Interval of mean difference,

[a] Between- group comparison of post intervention with analysis of covariance and effect sizes (ES) were calculated as Cohen d.

**Table 3. A backward stepwise logistic regression model with p values for each model variable predicting the main effects of prognostic variable (independent) on the outcome variable (post-intervention scores) at 6 weeks among the chronic LBP patients (n = 92) controlled for group allocation.**

| Outcomes (at 6 weeks) | Prognostic predictors at baseline | B | OR (95% CI) | $R^2$ | p-value for each model |
|---|---|---|---|---|---|
| Final disability (RMDQ$\geq$ 13/24) | Education status | 0.689 | 1.99 (1.42, 3.86) | 0.21 | 0.008 |
|  | Disability | -0.29 | 0.75 (0.41, 0.89) |  | 0.012 |
| Final pain ($\geq$ 5/10) | Pain intensity | 0.525 | 1.69 (1.31, 2.03) | 0.24 | 0.036 |
| Final WHO5 I ($\geq$ 60/100) | Age | 0.79 | 2.20 (1.47, 3.81) | 0.19 | 0.002 |
| Final GSES ($\geq$ 33/40) | Sex | 0.29 | 1.34 (1.09, 2.22) | 0.31 | <0.001 |
|  | Self-efficacy | 0.61 | 1.84 (1.46, 2.75) |  |  |

self-efficacy outcomes at 6 weeks were predicted by higher age (OR 2.20, 95% CI 1.47, 3.81), female gender (OR 1.34, 95% CI 1.09, 2.22), and higher self-efficacy at baseline (OR 1.84, 95% CI 1.46, 2.75).

## Discussion

The pain education intervention for chronic LBP patients yielded improved outcomes in terms of pain, disability, and well-being over a 6-week period, with effect sizes ranging from moderate to larger. The observed enhancements in pain, well-being, and disability may be attributed to the hypothesis that educating LBP patients in the experimental group about the nature of pain, particularly its association with central sensitization, and emphasizing movement over pain perception, facilitated a shift in patients' perceptions of LBP [40, 41]. This shift likely encouraged the adoption of a self-management approach incorporating bio-psychosocial principles, purportedly reducing LBP by diminishing central sensitivity [42, 43]. Additionally, higher self-reported disability and greater pain intensity emerged as indicators of poorer prognosis in this trial [44, 45].

Participants with higher educational attainment and females were more likely to report better well-being and self-efficacy after 6 weeks of intervention. This finding is unsurprising, as higher education may afford individuals greater insight into and comprehension of pain education programs, aiding in pain management and mitigating challenges associated with LBP [46, 47]. However, overall, significant improvements were observed in the primary outcomes (RMDQ and VAS) and well-being in the pain intervention group compared to the control group. Interestingly, no meaningful association was found in the logistic regression model between the prognosis of outcome measure scores and smoking habits, contrary to findings in some literature that identified smoking as a prognostic indicator in LBP patients [38, 44, 48]. Similar to the results of this study, a cognition-targeted intervention study [49] reported improvements in pain and disability among individuals with spinal pain. Their education-based intervention yielded larger effect sizes (0.66 and 0.81 for pain and disability, respectively). Likewise, in this study, the results at the end of the 6-week intervention showed effect sizes of 0.82 for pain and 0.75 for disability.

The effect sizes (ES) for pain and disability resulting from pain education interventions varied across trials. Malfliet et al. [17] reported a lower ES of 0.52 for pain and 0.49 for disability at a 3-month follow-up compared to this trial. In contrast, a pilot trial by Ibrahim et al. [50] reported a larger ES for disability (2.22) and pain (1.66) employing group patient education combined with motor control exercise as an intervention for low-resource rural community-dwelling adults with chronic LBP. The clinical change in Roland Morris Disability Questionnaire (RMDQ) scores demonstrated a substantial improvement in our experimental group, with a 54.15% decrease in disability levels compared to a 6.8% improvement in the control

group. Notably, an improvement of 30% or more in RMDQ score is rated as clinically relevant [51]. These findings further strengthen the claim on the effectiveness of the pain education intervention in reducing disability related to CLBP, which is consistent with studies reporting the efficacy of pain education interventions [13, 30, 47, 49]. Regarding pain intensity, as measured by the Visual Analog Scale (VAS), both the experimental and control groups showed a significant decrease following the intervention. However, the mean difference in pain intensity score change between the groups remained statistically significant after adjusting for confounding variables, with a larger effect size observed in the experimental group. This indicates that the pain education program led to a notable and statistically significant decline in pain intensity reported by participants in the experimental group at the 6-month post-intervention assessment.

Furthermore, the pain education intervention had a positive impact on the well-being of participants, as indicated by the WHO-5 well-being index. The experimental group in this trial demonstrated a significant improvement in well-being from baseline to the 6-month intervention period, with a moderate effect size. This positive effect on well-being remained consistent even after accounting for potential confounders, which is supported by evidence suggesting that psychological interventions are effective in improving well-being [52]. However, in this study, there were no significant differences between the experimental and control groups in terms of general self-efficacy. Further studies report an association of low self-efficacy among adults with CLBP and the need for interventions [53].

Pain education programs, when delivered alongside conventional treatment, have shown significant improvement in pain reduction compared to the other group [13, 17]. Similarly, two reviews [13, 54] have reported that pain education enhances pain reconceptualization, which might facilitate patients' ability to cope with their conditions. However, one of those reviews [54] found that pain education has only short-term effects when used alongside physiotherapy interventions in CLBP. Unfortunately, this study did not record the outcome measures at follow-up to investigate the retention effects of pain education intervention. The group receiving pain education has shown significant improvement, indicating that the comprehensive approach targeting both physical and psychological aspects of pain management offers promise for improving disability in individuals with chronic low back pain. Similar findings were reported by studies conducted by Saracoglu et al. (2020) and Malfliet et al. (2017) [17, 55], addressing that psychological and neurophysiological aspects of pain education enhance the understanding and management of pain, resulting in reduced pain levels, improved physical function, and decreased disability.

Pain education is reported to be effective in reducing self-reported disability and pain in chronic conditions like cancer, osteoarthritis, post-operative, and associated symptoms based on the findings of previous literature [56, 57]. The findings of this study also demonstrated that participants' quality of life and well-being improved with pain education. A better quality of life and an improved ability to engage in daily activities and pursue meaningful goals are needed for patients with chronic low back pain. The importance of educating patients about the underlying mechanisms of pain to improve their pain management and overall quality of life by pain education was emphasized by Louw and others [58]. The study by Moseley et al. (2014) highlighted the effectiveness of intensive neurophysiology education in reducing pain intensity and improving physical functioning and psychological well-being in individuals with chronic low back pain, thus promoting overall well-being [59]. Malfliet et al. also reported that quality of life can be improved by delivering pain education with motor control training in patients with low back pain [17]. Similarly, a cross-sectional study reported that pain self-efficacy accounts for a greater degree of variation in disability compared to fear of movement. Further, it was observed that changes in self-efficacy, rather than changes in fear of movement,

serve as a mediator between changes in pain intensity and changes in disability over the course of one year [60]. More importantly, pain education intervention can be tailored for individuals or to suit a particular group of individuals experiencing chronic pain. Interventions by pain education could be delivered outside the healthcare system at a very low resource set up. These characteristics of pain education suggest that it can be effective, versatile, and an alternative to current practice.

## Strength and limitations

This trial stands out as one of the few adequately powered studies in India to have been prospectively registered, featuring a design aimed at minimizing potential bias and providing insight into the potential changes that pain education can have on pain, disability, and well-being among chronic LBP patients. Additionally, the inclusion of a control group treated with a guidelines-based standardized plan of care distinguishes this study from others that may offer minimal or usual care control interventions. Despite its strengths, there are several limitations worthy of discussion that call for caution when interpreting the findings of this study. Firstly, India's linguistically and culturally diverse population may necessitate a tailored pain education program specific to the population, recognizing diverse needs for most behavioral treatment options. Secondly, being a single-center trial, the external validity may be limited, and caution should be exercised when generalizing the findings to a broader population. Furthermore, feasibility constraints prevented the recording of follow-up data, so the retention effect of the pain education program in this study remains unknown. Initial differences in RMDQ scores and smoking habits between the groups, as well as the experimental group receiving additional treatment, could potentially account for some of the findings. Lastly, although the fidelity of treatment delivery of pain education was monitored, resource constraints prevented a comprehensive evaluation of it.

## Conclusions

The findings of this study suggest that integrating a pain education program with standard physiotherapy care enhances therapeutic outcomes for individuals with chronic LBP. Based on these results, we conclude that pain education holds clinical benefits when administered alongside standard physiotherapy care in the management of chronic low back pain.

## Supporting information

**S1 File. SPIRIT protocol of the trial.**
(DOCX)

**S2 File. Pain education manual English & Hindi language version 1.0.**
(PDF)

**S3 File. LBP RCT dataset.**
(XLS)

## Author Contributions

**Conceptualization:** Mohammad Sidiq, Tufail Muzaffar, Shariq Masoodi, Arunachalam Ramachandran, Aksh Chahal, Moattar Raza Rizvi.

**Data curation:** Balamurugan Janakiraman, Rajkumar Krishnan Vasanthi.

**Formal analysis:** Balamurugan Janakiraman, Faizan Zaffar Kashoo.

**Investigation:** Moattar Raza Rizvi, Ankita Sharma, Rituraj Verma, Monika Sharma, Sajjad Alam, Krishna Reddy Vajrala, Jyoti Sharma.

**Methodology:** Mohammad Sidiq, Nitesh Bansal, Ankita Sharma, Sajjad Alam, Krishna Reddy Vajrala.

**Project administration:** Aksh Chahal.

**Supervision:** Shariq Masoodi, Richa Hirendra Rai.

**Visualization:** Richa Hirendra Rai.

**Writing – original draft:** Balamurugan Janakiraman, Ramprasad Muthukrishnan.

**Writing – review & editing:** Balamurugan Janakiraman.

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
