## [Decision Letter · Decision Letter 0]

22 Jan 2024

**PONE-D-23-35122**

Effectiveness of pain education on pain, disability, quality of life and self-efficacy in chronic low back pain: A randomized controlled trial

*PLOS ONE*

Dear Dr. Muthukrishnan, I hope this finds you well.

First of all, thank you for submitting your manuscript to PLOS ONE. I would like to congratulate you on your research. You aimed to gain an understanding of the effectiveness of pain education on pain levels, disability, quality of life, and self-efficacy in individuals with chronic low back pain. I kindly ask you to adjust the article according to my requests and the requests of the three reviewers. In this review, please mark your corrections and report the page and adjustment line. Besides, please:

—Change the title to: Effects of pain education on disability, pain, quality of life, and self-efficacy in chronic low back pain: A randomized controlled trial.

—Include keywords according to MeSH terms (https://www.ncbi.nlm.nih.gov/mesh/);

—The objective stated in your abstract is different from the objective stated in your introduction. Please correct this. The sentence describing the objective must be exactly the same in both;

—State a hypothesis in the introduction;

—The conclusion described in the abstract and the conclusion described after the discussion must be exactly the same;

—In the Methods (Data analysis) section, describe the classification used for the effect size (Cohen's d [https://pubmed.ncbi.nlm.nih.gov/37971135/]), since you reported it as n2 in Table 2.

—In Table 1, report the body mass (kg) and stature (cm);

—In Table 2, remove the description of within-group comparisons (not necessary for this RCT). Notice that your research question is answered by between-group comparisons.

—In Table 3, change p = 0.000 to p <0.001;

—Submit the database with the data in unanalyzed values (i.e., absolute values) as a supplemental file (Excel and English).

We look forward to receiving your revised manuscript.

Kind regards,

André Pontes-Silva

Academic Editor

PLOS ONE

2. In the online submission form, you indicated that [All the data supporting the findings of this trial is presented and the full dataset are available from the principal investigator on request.]. 

3. Please amend your authorship list in your manuscript file to include author Alagappan Thiyagarajan.

Reviewers' comments:

Reviewer's Responses to Questions

**Comments to the Author**

1. Is the manuscript technically sound, and do the data support the conclusions?

Reviewer #1: Partly

Reviewer #2: No

Reviewer #3: Partly

2. Has the statistical analysis been performed appropriately and rigorously? 

Reviewer #1: Yes

Reviewer #2: No

Reviewer #3: No

3. Have the authors made all data underlying the findings in their manuscript fully available?

Reviewer #1: Yes

Reviewer #2: Yes

Reviewer #3: No

4. Is the manuscript presented in an intelligible fashion and written in standard English?

Reviewer #1: Yes

Reviewer #2: Yes

Reviewer #3: Yes

5. Review Comments to the Author

Reviewer #1: Thank you for the opportunity to review this article.

Below is my analysis of the manuscript.

Overall impression:

The study presents a solid scientific basis. There is significant thematic relevance and remarkable coherence in giving the context of previous literature, especially when considering the distinct cultural differences in each continent.

Main issues:

1) According to the journal's guidelines, including the trial or study's registration number in the abstract is essential.

2) The last paragraph of the introduction could be more concise, avoiding excessive repetition of terms.

3) Improving the description of the standard physiotherapeutic treatment is essential. It would be relevant to provide details of the physical assessments carried out and clarify whether all patients received the same type of treatment regarding sets, repetitions and weights. This is especially crucial considering the diversity in the fitness levels of the participating patients.

4) When writing the conclusion, it is important to exercise caution to ensure conciseness and a clear answer to the question raised in the introduction. The decision should be aligned with the results obtained, avoiding inappropriate wording.

Reviewer #2: This is an interesting study that is looking at the effect of pain education on pain intensity. However would benefit from some proof reading, as they are spelling mistakes in some places.

Major observation:

Primary outcomes measures are: The primary outcomes measures were related to the intensity of pain experienced during activities of daily life (ADL) and disability related to pain. Roland Morris Disability Questionnaire (RMDQ). The sample size is based on "mean score of pain intensity in chronic LBP patients (mean1 = 3.76, mean2 = 4.78, SD1 = 1.51, SD2 = + 1.91)". Its unclear whether this is a combination of of the two ADL and RMDQ - which by definition are co-primary endpoints. This needs to be reflected in the sample size, and the results as its confusing, there is reduction in disability and pain intensity.

In addition, if the question was addressing two co-primary endpoints? Then more information is required in terms multiplicity testing or not and why? Or pre-specified interpretations in terms of whether intervention works if one or both or either outcomes are observed to be clinically significant.

Some further suggestions;

1. In the introduction section, mention that the primary was at 6 weeks.

2. With the primary outcome, can the authors include the collection time points, i.e. in the data analysis section, the authors mention, post intervention. Is this at 6 weeks, or at follow-up time point?

3. As this is a randomised controlled trial, there needs to be distinction in terms of the analysis to reflect this. Its not clear what the main objective was.

Suggested to update the data analysis section,

-To include that as this an RCT, this will be reported in accordance to the CONSORT guidelines?

4. Why was lost to follow-up not accounted for in the sample size, or include a sentence to justify why not, i.e. by design because the intervention last 6 weeks, so were the researchers confident enough that participants would adhere?

5. It is not recommended to test baseline characteristics, any differences between baseline characteristics would be due to chance. - see this interesting article Correct Baseline Comparisons in a Randomized Trial Schober, Patrick MD, PhD, MMedStat*; Vetter, Thomas R. MD, MPH†

"Traditionally, baseline balance is assessed by a series of hypothesis tests comparing each baseline variable between the groups, and a nonsignificant result (P > .05) has commonly been considered to indicate baseline balance. This approach is flawed.2 First, all patients are randomly sampled from the same population, and any differences must be due to chance. It thus does not make sense to test hypotheses of group differences at baseline. Second, hypothesis tests are greatly affected by sample size. In a smaller trial, marked differences between the groups can be nonsignificant, whereas in a larger trial, negligible differences could be significant."

6. This sentence is incomplete "Unadjusted mean (SD), and standard error (SE) of the mean difference (MD) between the groups with associated 95% confidence interval and p-value were computed for the outcome measures at baseline and post-intervention"- using what statistical approach/model?

7. The authors jump straight to logistic prognostic model - which is not clear whether this was separate objective. The most appropriate approach would be better to be explicit as to whether the outcomes were analysed with the pre-specified baseline characteristics for assessing the treatment effect.

The statistical analysis of prognostic model should be differentiated from the research question (primary) and secondary (which are questions assessing the effect of the intervention) - otherwise its confusing.

8. How was missing data handled- this is not mentioned?

9. Also in the statistical analysis section, mention when the SAP was drafted etc.

10. How was adherence/compliance assessed

11. Table 1 - add percentage for gender (male and female).

12. There is imbalance in terms of occupation, interestingly more housewives in the control group.

13. Additional information is missing in the randomisation section include allocation ratio.

14. Figure 2 - put units on y-axis (e.g mean score difference

15. Discuss limitation in terms of generalizability, as this was in one single centre?

Reviewer #3: I appreciate the opportunity to review this study. The authors conducted a randomized controlled trial aiming to assess the effectiveness of pain education on pain levels, disability, quality of life, and self-efficacy in individuals with chronic low back pain (CLBP). The manuscript is well-written and detailed, indicating a significant investment of time and effort. I will provide specific comments on the manuscript below.

Abstract:

-  I recommend revising the aim as follows: “This randomized controlled trial aimed to evaluate the effectiveness of pain education on pain levels, disability, quality of life, and self-efficacy in individuals with chronic low back pain (CLBP)”.

Introduction:

- It is crucial to include references to previous studies that have assessed the effect of PNE in patients with CLBP. Provide a summary of their findings and highlight the novelty of the present study.

- The introduction should include the scientific background of the topic and an explanation of the rationale.

The authors need to clearly state the hypothesis of the study.

Methods:

Reference “Figure 1” as the flow diagram in the manuscript.

If there were any modifications to the methods after trial registration, specify them.

Provide a detailed description of the intervention called “standard physiotherapy." It would be important not only for transparency but also to offer valuable information for future studies.

Assess and report on the blinding procedures. How can the authors ensure that appropriate blinding is achieved?

Sample size:

Recommend recalculating the sample size, especially considering the Roland-Morris Disability Questionnaire (RMDQ) as a primary outcome. Address the baseline differences between groups, as comparing groups with distinct profiles is a serious concern.

Examine the differences between groups for specific variables like smokers and RMDQ scores. Clarify how these differences were managed in the analysis.

Discuss how the authors addressed the potential bias introduced by one group receiving more treatment, potentially influencing their improvement compared to the control group. Acknowledge this as a potential limitation.

Prognostic model:

Clarify why the prognostic model was not explicitly stated as the aim of the study.

6. PLOS authors have the option to publish the peer review history of their article (what does this mean?). If published, this will include your full peer review and any attached files.

Reviewer #1: No

Reviewer #2: No

Reviewer #3: **Yes: **

---

## [Author Response · Author response to Decision Letter 0]

10 Feb 2024

Dear Editor, the responses are attached as a word file in point by point manner. We are grateful to the learned reviewers, who took their precious time in evaluating and appraising our manuscript, and all the necessary changes and recommendations have been incorporated accordingly in a file named " author responses to reviewers".

---

## [Decision Letter · Decision Letter 1]

8 Apr 2024

PONE-D-23-35122R1Effects of pain education on disability, pain, quality of life, and self-efficacy in chronic low back pain: A randomized controlled trialPLOS ONE

Dear Dr. Muthukrishnan, I hope you are well.

Please review your article and submit it for final consideration.

Minor revisions:

1- Sample Size Calculation: Indicate the statistical testing method used for the sample size calculation. Perhaps it is the t-test.

2- Data analysis section:

a. Typographical error: Student’s t-test.

b. State the underlying covariance structure used in the linear mixed model and the criteria for selecting it.

3- In addition to the mean age, include the corresponding standard deviation. Throughout the manuscript when stating means, provide the corresponding standard deviations.

4- Table 1: In the statistical methods section, state and describe the use of all statistical methods.

a. Indicate the statistical testing method used for comparing categorical data.

b. State and describe the use of logistic regression modeling.

5- To assist in the review process, add line numbering to the document.Indicate which changes you require for acceptance versus which changes you recommendAddress any conflicts between the reviews so that it's clear which advice the authors should followProvide specific feedback from your evaluation of the manuscriptPlease ensure that your decision is justified on PLOS ONE’s publication criteria and not, for example, on novelty or perceived impact.

We look forward to receiving your revised manuscript.

Kind regards,

André Pontes-Silva

Academic Editor

PLOS ONE

Journal Requirements:

Reviewers' comments:

Reviewer's Responses to Questions

**Comments to the Author**

1. If the authors have adequately addressed your comments raised in a previous round of review and you feel that this manuscript is now acceptable for publication, you may indicate that here to bypass the “Comments to the Author” section, enter your conflict of interest statement in the “Confidential to Editor” section, and submit your "Accept" recommendation.

Reviewer #1: (No Response)

Reviewer #4: (No Response)

2. Is the manuscript technically sound, and do the data support the conclusions?

Reviewer #1: Yes

Reviewer #4: Yes

3. Has the statistical analysis been performed appropriately and rigorously? 

Reviewer #1: Yes

Reviewer #4: Yes

4. Have the authors made all data underlying the findings in their manuscript fully available?

Reviewer #1: Yes

Reviewer #4: Yes

5. Is the manuscript presented in an intelligible fashion and written in standard English?

Reviewer #1: Yes

Reviewer #4: Yes

6. Review Comments to the Author

Reviewer #1: Thank them for promptly responding to the suggested corrections and demonstrating their commitment. The adjustments made to the sections, especially the methodology, contributed significantly to strengthening the argument and clarity of the text.

I encourage the authors to consider the observations made for future research.

Reviewer #4: This two-arm randomized controlled clinical trial aimed to assess the effects of pain education on pain levels, disability, quality of life, self-efficacy, and prognostic characteristics in individuals with chronic low back pain as assessed before and after 6-weeks of intervention. Post-intervention scores significantly differed at post intervention between the groups in the areas of disability, pain intensity, and well-being index, where the intervention group had better results.

Minor revisions:

1- Sample Size Calculation: Indicate the statistical testing method used for the sample size calculation. Perhaps it is the t-test.

2- Data analysis section:

a. Typographical error: Student’s t-test.

b. State the underlying covariance structure used in the linear mixed model and the criteria for selecting it.

3- In addition to the mean age, include the corresponding standard deviation. Throughout the manuscript when stating means, provide the corresponding standard deviations.

4- Table 1: In the statistical methods section, state and describe the use of all statistical methods.

a. Indicate the statistical testing method used for comparing categorical data.

b. State and describe the use of logistic regression modeling.

5- To assist in the review process, add line numbering to the document.

7. PLOS authors have the option to publish the peer review history of their article (what does this mean?). If published, this will include your full peer review and any attached files.

Reviewer #1: No

Reviewer #4: No

---

## [Author Response · Author response to Decision Letter 1]

9 Apr 2024

Thank you for the suggestions and we have amended the minor corrections recommended.

---

## [Editor Report · Decision Letter 2]

16 Apr 2024

Effects of pain education on disability, pain, quality of life, and self-efficacy in chronic low back pain: A randomized controlled trial

PONE-D-23-35122R2

Dear Dr. %Muthukrishnan%, I would like to thank you for your responses and corrections. 

We’re pleased to inform you that your manuscript has been judged scientifically suitable for publication and will be formally accepted for publication once it meets all outstanding technical requirements. 

Kind regards,

André Pontes-Silva

Academic Editor

PLOS ONE

---

## [Editor Report · Acceptance letter]

7 May 2024

PONE-D-23-35122R2 

PLOS ONE

Dear Dr. Muthukrishnan, 

I'm pleased to inform you that your manuscript has been deemed suitable for publication in PLOS ONE. Congratulations! Your manuscript is now being handed over to our production team.

Kind regards, 

on behalf of

Professor André Pontes-Silva 

Academic Editor

PLOS ONE